# Centroid- and Orientation-aware Feature Learning

## Abstract

Robust techniques for learning centroids and orientations of objects and shapes in two-dimensional images, along with other features is crucial for image- and video-processing applications. While this has been partially addressed using a number of techniques by achieving translational and rotational equivariance and invariance properties, learning them as part of the features still remains an open problem. In this paper, we propose a novel encoder-decoder-based mechanism for learning independent factors of variations, including centroids and orientations, by embedding special layers to achieve translational and rotational equivariance and invariance. Our evaluation, across a number of datasets, including that of real-world ones, against five different state-of-the-art baseline models shows that our model not only can offer superior disentangling and reconstruction performance, but also offers exceptional training and inference performance, as much as $10\times$ for training and $9\times$ on inference compared to the average performance of other models.

## 1 Introduction

Robust learning of centroids and orientations of objects in images, along with other features, underpins a number of downstream tasks, such as object detection, image registration, image classification and image or video stabilization (Jaderberg et al., 2015; Wang et al., 2018), with practical utility of this spanning into applications in scientific domains, such as protein and galaxy studies (Dieleman et al., 2015; Bendory et al., 2020). However, to robustly learn these intrinsic properties, the underlying neural network architecture should have the notion of translational and rotational equivariance and invariance, so that variations in locations and orientations do not affect the learning.

Several approaches have been proposed in the literature for robust learning of these features (Cohen et al., 2018; Tai et al., 2019; Cobb et al., 2020). These approaches focus on embedding inductive bias into neural networks (NNs), for example, via data augmentation, parameter constraints, explicitly and implicitly embedding equivariance constraints, or designing very bespoke loss functions (Jaderberg et al., 2015; Dieleman et al., 2016; Marcos et al., 2017; Sabour et al., 2017; Shorten & Khoshgoftaar, 2019). The advent of convolutional neural networks (CNNs) provided an important impetus to these efforts by partially embodying translational equivariance and invariance (Bietti & Mairal, 2019). However, CNNs are not naturally equivariant to other transformations, such as rotation. These shortcomings were addressed by using group convolutional layers (Cohen & Welling, 2016; Weiler et al., 2018). The main concept here is to design convolutional layers that exhibit equivariance with respect to a specific group of transformations. Having group pooling layers after the group convolutional layers enables the network to achieve invariance under these transformations.

While designing models to learn the centroids and orientations is useful, in many of the cases, these two features are two of the many features that meant to be learned in images. In this regard, approaches to disentanglement, especially in an unsupervised manner, are more appealing as they can offer an avenue for simultaneously learning all relevant features, including centroids and orientations. To handle this, approaches underpinned by bottleneck architecture models, where the features in a lower dimensional space are enforced to be independent or orthogonal, are directly relevant (Higgins et al., 2016; Cha & Thiyagalingam, 2023). While these models hold a great promise for disentangling features, they also inherently rely on the ability of the model to understand symmetry. A number of approaches have been proposed in the literature to overcome these issues. For

example, a large body of work employs grid sampling in the latent space, explicitly incorporating centroid and orientation information into the decoder (Bepler et al., 2019; Nasiri & Bepler, 2022; Kwon et al., 2023). While grid sampling is effective enough to provide pseudo centroids and orientations, they suffer from many issues, including discretization artifacts and computational complexity.

In this paper, by building on the notion of disentanglement, we propose a novel unsupervised approach to learn centroid and orientation information as part of other features. Underpinned by relevant theoretical foundations for establishing both translational and rotational equivariant and invariant layers inside an encoder architecture, we overcome the limitations of existing approaches. To this end, we propose CODAE, Centroid- and Orientation-aware Disentangling Autoencoder, with the following key contributions:

- we design an encoder with two arms: one arm specializes in learning the translational and rotational equivariant features, while the other arm focuses on their invariant counterpart,

- we propose an approach, namely, use of image moments, to overcome the potential issues of discretization arising from the translational and rotational equivariant layers,

- we eliminate the need for additional loss functions, and keep the number of loss functions to a minimum, and

- we demonstrate that the proposed model offers superior training and inference performance when compared to the other models, making it more amenable to a range of practical cases.

Our evaluation over synthetic and realistic datasets shows that the proposed model outperforms similar approaches and offers state-of-the-art results in many tasks.

## 2 RELATED WORK

Learning centroids and orientations of shapes along with other features is a rather common problem in machine learning, and as such, the literature is considerably rich. Here, in this work, we are focussing on an unsupervised approach that is reliant on the group $SO(2)$ for the two-dimensional case. Therefore, we limit our related work analysis to the work that are directly relevant here, which we outline in the following subsections.

### 2.1 TRANSLATIONAL AND ROTATIONAL EQUIVARIANT NEURAL NETWORKS

Data augmentation was a primary technique, especially at the early stages of NN-based designs, to achieve translational and rotational equivariant representations (Shorten & Khoshgoftaar, 2019). The design of CNNs were instrumental for achieving translational equivariance, and significant efforts have been made to develop layers that are add-on to CNN model that are equivariant to both translation and rotation (Cohen & Welling, 2016; Weiler et al., 2018). Unlike data augmentation, these efforts centered around layer operations. To make NNs equivariant to symmetry groups, such as permutation and rotation groups, Cohen & Welling (2016) proposed steerable dense convolution layers. The filters in the layers can be steered to capture features based on symmetry transformations, aligning with the properties of a symmetry group. By using steerable filters, the layers can reduce the number of parameters needed to achieve similar performance compared to using a larger number of fixed filters, offering a more effective feature extraction mechanism. Furthermore, a group pooling layer, that takes the maximum activation within its inputs, achieves invariant features. By selecting the appropriate symmetry group, such as $SO(2)$, or cyclic group, two-dimensional rotational equivariance and invariance can be achieved.

### 2.2 DISENTANGLED REPRESENTATION LEARNING

The objective of disentangled representation learning is to learn a set of independent factors, representing independent variations from one another while representing the entire dataset (Higgins et al., 2018). Generative models, especially those built upon variational autoencoders (VAEs), have shown their effectiveness in learning disentangled representations (Higgins et al., 2017). Numerous models based on VAEs have been developed, introducing novel loss functions that enforce learning of independent features in the latent space. However, with a formal definition of disentanglement through

the mathematical concepts of group theory, coupled with limitation of generative models in learning disentangled representations outlined in Locatello et al. (2019), new approaches have emerged to build based on the principles of symmetry transformations. Their aim is to discover transformations within the latent space that remain independent of one another, underpinned by the mathematical frameworks (Tonnaer et al., 2022; Cha & Thiyagalingam, 2023). In addition to the architecture design, there have been numerous metrics proposed to quantitatively assess the degree of disentanglement. In our study, we adopt the notions introduced in Zaidi et al. (2020), where the metrics are divided into three classes based on their fundamental approaches, namely, Intervention-based, Predictor-based, and Information-based metrics.

## 2.3 APPROACHES TO LEARNING CENTROIDS AND ORIENTATIONS

Significant attention has been dedicated to separately learn centroids and orientations of images, enabling to reconstruct aligned images. The Spatial-VAE, which is based on VAE, utilizes a spatial decoder, that maps spatial coordinates to pixel values (Bepler et al., 2019). As it generates pixel values for every coordinate, it uses a multilayer perceptron (MLP) – an approach that is computationally expensive. While this model can partially reconstruct aligned images for certain datasets, such as proteins and galaxies, the approach fails on some, such as the MNIST dataset, owing to the symmetry issues when digits are rotated through angles ranging between $-180°$ and $180°$. The Target-VAE addresses this issue by designing a group convolution layers-based encoder that outputs attention values, angles, and content components (Nasiri & Bepler, 2022; Cesa et al., 2021). The attention values assist in learning rotation angles from the discretized rotation component, effectively resolving the issue in the Spatial-VAE. Considering the performance of the spatial decoder in the two models, another approach, namely, IRL-INR was proposed in Kwon et al. (2023). This approach maps the spatial coordinates to pixel values through a hypernetwork decoder with two additional loss functions: i) consistency loss, and ii) symmetry-breaking loss. The former uses pairs of images to learn invariant features, while the latter uses the centroids and orientations obtained from the first-order image moment to learn those by the model. While this model produces high-resolution aligned images, it requires a larger latent dimension compared to the other models due to the decoder design. In overall, all these models have computationally expensive decoder, and their primary focus is not on achieving disentanglement but rather on obtaining centroids and orientations.

## 3 FRAMEWORK

In this section, we define layers that are translational and rotational equivariant and invariant. In addition to this, we introduce image moments that complement the limitations arising from discretization in the implementation of the proposed layer.

## 3.1 TRANSLATIONAL AND ROTATIONAL EQUIVARIANT LAYERS

It is well known that the conventional two-dimensional convolutional layer inherently possesses translational equivariance. To obtain equivariance in both translation and rotation, we introduce supplementary operations to the conventional convolution process. In the sequel, we consider functions $f \in L^p(\mathbb{R}^2)$ and $g \in L^q(\mathbb{R}^2)$, $\frac{1}{p} + \frac{1}{q} = 1 (p \geq 1)$, so that $\int_{\mathbb{R}^2} f(\boldsymbol{x})g(\boldsymbol{x})\, d\boldsymbol{x} < \infty$.

**Theorem 3.1.** *For functions* $f \in L^p(\mathbb{R}^2)$ *and* $g \in L^q(\mathbb{R}^2)$*, let* $L_{\boldsymbol{t}}$ *be a function defined by* $(L_{\boldsymbol{t}}f)(\boldsymbol{x}) = f(\boldsymbol{x} - \boldsymbol{t})$*, for* $\boldsymbol{x}, \boldsymbol{t} \in \mathbb{R}^2$*. Then the 2D-convolution of* $L_{\boldsymbol{t}}f$ *and* $g$*, denoting the operator by the symbol* $*$*, is translational equivariant (Cohen & Welling, 2016).*

Theorem 3.1 shows that the conventional two-dimensional convolution already satisfies translational equivariance property. We now introduce supplementary operations using the convolution above to obtain rotational equivariance. In order to have an operation which is rotational equivariance, we consider a function $\tilde{L}_\phi$ exhibiting rotation by $(\tilde{L}_\phi f)(\boldsymbol{x}) = f(R_{-\phi}\boldsymbol{x})$ for $\boldsymbol{x} \in \mathbb{R}^2$ and $\phi \in \mathbb{R}$, where

$$R_\phi = \left[ \begin{array}{cc} \cos(\phi) & -\sin(\phi) \\ \sin(\phi) & \cos(\phi) \end{array} \right]. \tag{1}$$

**Theorem 3.2.** *Let* $\tilde{*}$ *be an operation defined by* $(f\tilde{*}g)(\boldsymbol{x}) = \int_{S^1} \int_{\mathbb{R}^2} f(\boldsymbol{y})g(R_{-\theta}(\boldsymbol{x} - \boldsymbol{y}))\, d\boldsymbol{y}\, d\theta$*. Then* $\tilde{*}$ *is equivariant under both translations and rotations.*

*Proof.* (a) From Theorem 3.1, it is easy to see that $\tilde{*}$ is translational equivariant.

(b) For $\boldsymbol{x} \in \mathbb{R}^2$ and $\psi \in \mathbb{R}$, we obtain that

$$[(\tilde{L}_\phi f)\tilde{*}g](\boldsymbol{x}) = \int_{S^1} \int_{\mathbb{R}^2} (L_\phi f)(\boldsymbol{y})g(R_{-\theta}(\boldsymbol{x} - \boldsymbol{y}))\, d\boldsymbol{y}\, d\theta \tag{2}$$

$$= \int_{S^1} \int_{\mathbb{R}^2} f(R_{-\phi}\boldsymbol{y})g(R_{-\theta}(\boldsymbol{x} - \boldsymbol{y}))\, d\boldsymbol{y}\, d\theta \tag{3}$$

$$= \int_{S^1} \int_{\mathbb{R}^2} f(\tilde{\boldsymbol{y}})g(R_{-\theta}(\boldsymbol{x} - R_\phi\tilde{\boldsymbol{y}}))\, d\tilde{\boldsymbol{y}}\, d\theta \tag{4}$$

$$= \int_{S^1} \int_{\mathbb{R}^2} f(\tilde{\boldsymbol{y}})g(R_{-\theta}R_\phi(R_{-\phi}\boldsymbol{x} - \tilde{\boldsymbol{y}}))\, d\tilde{\boldsymbol{y}}\, d\theta \tag{5}$$

$$= \int_{S^1} \int_{\mathbb{R}^2} f(\tilde{\boldsymbol{y}})g(R_{-\tilde{\theta}}(R_{-\phi}\boldsymbol{x} - \tilde{\boldsymbol{y}}))\, d\tilde{\boldsymbol{y}}\, d\tilde{\theta} \tag{6}$$

$$= (f\tilde{*}g)(R_{-\phi}\boldsymbol{x}) = \tilde{L}_\phi(f\tilde{*}g)(\boldsymbol{x}), \tag{7}$$

where $\tilde{\boldsymbol{y}} = R_{-\phi}(\boldsymbol{y})$ and $\tilde{\theta} = \theta - \phi$.

$\square$

**Theorem 3.3.** *Let $f$ be a function on $\mathbb{R}^2$ such that $\sup_{\boldsymbol{x}\in\mathbb{R}^2} f(\boldsymbol{x}) < \infty$. Then the supremum of the function is invariant under both translations and rotations.*

*Proof.* Let $M = \sup_{\boldsymbol{x}\in\mathbb{R}^2} f(\boldsymbol{x})$. Then for any $\boldsymbol{t} \in \mathbb{R}^2$ and $\phi \in \mathbb{R}$, it is easy to check that $M = \sup_{\boldsymbol{x}\in\mathbb{R}^2} f(\boldsymbol{x} - \boldsymbol{t}) = \sup_{\boldsymbol{x}\in\mathbb{R}^2} f(R_\phi\boldsymbol{x})$. Hence $M$ is invariant under translations and rotations. $\square$

We note that $\sup_{\boldsymbol{x}\in\mathbb{R}^2} f(\boldsymbol{x}) = \max_{\boldsymbol{x}\in\mathbb{R}^2} f(\boldsymbol{x})$ if $f$ has a continuous function which has a compact support, i.e. the closure of $\{\boldsymbol{x} \in \mathbb{R}^2 : f(\boldsymbol{x}) \neq 0\}$ is closed and bounded.

## 3.2 IMAGE MOMENTS

While the concept of moments was initially introduced in the context of statistics, it has been widely applied in the field of computer vision (Hu, 1962; Prokop & Reeves, 1992; Flusser, 2006). Moments, which originally describe distribution of data points, are capable of capturing essential features and characteristics of images, including the center of mass and orientation of objects. Here, we present the relevant background regarding image moments. Let $f(x, y)$ be a 2D-continuous function. Then the moment of order $(p + q)$ is defined as

$$M_{pq} = \int_{-\infty}^{\infty} \int_{-\infty}^{\infty} x^p y^q f(x, y)\, dx\, dy \tag{8}$$

for $p, q \in \mathbb{Z}^+ \cup \{0\}$. For a gray-scale image $I$, $M_{00}$ represents an area of $I$, and the centroid of $I$ can be represented by the first order moments as $(\bar{x}, \bar{y}) = \left(\frac{M_{10}}{M_{00}}, \frac{M_{01}}{M_{00}}\right)$. Furthermore, in order to describe the shape of the function without considering translation, we can use central moments defined as

$$\mu_{pq} = \int_{-\infty}^{\infty} \int_{-\infty}^{\infty} (x - \bar{x})^p (y - \bar{y})^q f(x, y)\, dx\, dy. \tag{9}$$

Then the second order central moments are used to extract the orientation of $I$. First, we calculate the covariance matrix of $I$,

$$Cov(I) = \begin{bmatrix} c_{20} & c_{11} \\ c_{11} & c_{02} \end{bmatrix}. \tag{10}$$

where $c_{ij} = \frac{\mu_{ij}}{\mu_{00}}$. Since the eigenvectors of this matrix correspond to the directions of maximum and minimum variance of image intensity, the orientation can be extracted from the angle formed between the eigenvector associated with the largest eigenvalue and the axis that is closest to this eigenvector. Therefore, the orientation $\alpha$ can be expressed by

$$\alpha = \frac{1}{2} \arctan\left(\frac{2c_{11}}{c_{20} - c_{02}}\right). \tag{11}$$

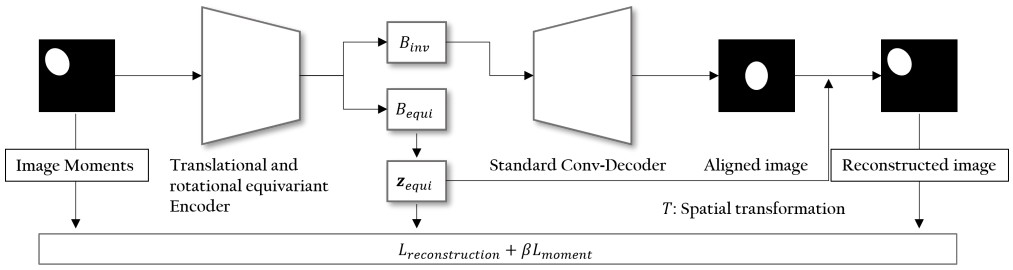

Figure 1: Illustration of the CODAE architecture. The model includes two arms and the spatial transformation to effectively learn centroid and orientation along with other features.

However, this orientation only captures the angle formed between the eigenvector associated with the largest eigenvalue and the axis that is closest to this eigenvector. Therefore, two images which are flipped by any line that passes through the center will have the same orientation value as discussed in Prokop & Reeves (1992). To address this issue, Prokop & Reeves (1992) further used the third order moments to capture the skewness of the image. The skewness represents how the pixel intensities are distributed asymmetrically around the mean intensity. This value distinguishes between two mirror images, which exhibit symmetry and consequently possess opposite skewness signs. In this study, we firstly align the image, $I$, with the eigenvector using the value $\alpha$ in Eq. 11 to obtain an aligned image, denoted by $\tilde{I}$. Then, one of the third order moments of $\tilde{I}$, which we denote $\tilde{M}_{30}$, is calculated to describe the skewness of $\tilde{I}$ along the $x$-axis. Since the rotation of an image by $180°$ changes the sign of the skewness of the projection on the x-axis, it can distinguish the two mirror images. Therefore, we assign the orientation of $I$ by

$$\bar{\alpha} = \begin{cases} \alpha, & \text{if } \tilde{M}_{30} > 0 \\ \alpha + \pi, & \text{otherwise.} \end{cases} \tag{12}$$

### 3.3 Network Architecture

Our main goal is to learn meaningful features along with centroid and orientation. To achieve this, we use DAE as our base model (Cha & Thiyagalingam, 2023). The advantage here is that the DAE, unlike most VAE-based models, relies purely on the reconstruction loss. Upon relying on the DAE model, we utilize translational and rotational equivariant layers in the Encoder part, which is denoted by $E_\phi$. This allows extracting meaningful features while retaining translation and rotation information of the input images. The translational and rotational equivariant layers are implemented based on Cesa et al. (2021), primarily for the reasons of seeking an efficient implementation.

To independently learn centroid and orientation along with other features, we design two branches at the end of the translational and rotational equivariant layers. The first branch, denoted by $B_{inv}$, is followed by a max pooling layer, that enables learning translational and rotational invariant features as described in Theorem 3.3. These features are then input to the decoder, which is denoted by $D_\theta$, to output images that are aligned with each other. The second branch, denoted by $B_{equi}$, learns centroid and orientation factors by explicitly utilizing these factors to reconstruct the input images. An affine transformation is used to translate and rotate the aligned images within the proposed model.

In addition to the reconstruction loss between input and reconstructed images, we introduce a loss to guide the centroid and orientation feature learning using image moments. Although image moments capture the centroid and orientation features with a high degree of accuracy, they still have a degree of subtle inaccuracy. Therefore, we utilize the image moments as guiding principles during the initial stages of learning. Consequently, our loss function can be written as:

$$L_r(I, T_{\boldsymbol{z}_{equi}}(D_{\boldsymbol{\theta}}(\boldsymbol{z}_{inv}))) + \beta L_m(\{\bar{x}, \bar{y}, \bar{\alpha}\}, \boldsymbol{z}_{equi}), \tag{13}$$

where $\boldsymbol{z}_{equi}$ and $\boldsymbol{z}_{inv}$ are the outputs of $B_{equi}$ and $B_{inv}$, respectively, and $T_{\boldsymbol{z}_{equi}}$ is an affine transformation using $\boldsymbol{z}_{equi}$. While the $L_r$ loss compares the input images with the reconstructed images, the $L_m$ loss compares the image moments with $\boldsymbol{z}_{equi}$. The value of $\beta$ is gradually decreased from a large value to zero, thereby guiding the model to learn the centroid and orientation features during its initial learning stage. The model framework can be shown in Figure 1.

## 4 EVALUATION AND RESULTS

Our evaluation involves comparing the performance of the proposed approach against five baseline models across six datasets. The code will be publicly available when the paper is published. We outline the evaluation details below.

**Datasets:** The 5HDB, MNISTU, XYRCS, and dSprites datasets contain reliable ground truth labels (Deng, 2012; Matthey et al., 2017; Lin et al., 2016; Bepler et al., 2019). The MNISTU dataset is derived from the MNIST dataset with random rotation using angles uniformly distributed between $-180°$ and $180°$. The XYRCS is a simple yet effective synthetic dataset containing three shapes (a circle, a triangle and a rectangle) with varying $x$ and $y$ positions, orientations and color information (more specifically, the brightness). Among four datasets, only XYRCS and dSprites datasets contain all ground truth factors while limited ground truth factors are available for the MNISTU and 5HDB datasets. Hence, we use the XYRCS and dSprites datasets for comparing the disentanglement capability of different models with and without $x$ and $y$ positions and orientations. This offers an approach for evaluating the capability of different models to learn invariant features along with centroids and orientations. Finally, in addition to these four datasets, we also use two real-world datasets from two different scientific domains, namely, EMPIAR-10029 (from life sciences) and Galaxy-zoo (from astronomy) datasets to demonstrate the performance of the proposed approach on realistic and complex datasets (Iudin et al., 2023; Lintott et al., 2008).

**Baseline Models:** We use the DAE (Cha & Thiyagalingam, 2023), $\beta$-VAE (Higgins et al., 2016), Spatial-VAE (Bepler et al., 2019), Target-VAE (Nasiri & Bepler, 2022), and IRL-INR (Kwon et al., 2023) as our baseline models. We selected these as baseline models in an unbiased manner, purely based on the approach as outlined in Section 2.

**Metrics of Evaluation:** We use two types of metrics: **(a) Numerical scores**, and **(b) Visualization of aligned images from input images and the reconstructions of latent traversals**. The former quantifies capability of models to either learn features for downstream tasks or independent features. We use the 5HDB, MNISTU, XYRCS, and dSprites datasets, that fully or partially contain ground truth factors, as stated above. Firstly, the 5HDB dataset includes the ground truth rotation angles. Therefore, we have calculated the Pearson Correlation Coefficient (PCC) between the actual rotation angles and those predicted by models. Secondly, given the ground truth labels in the MNISTU dataset, we have evaluated downstream task performances using Silhouette score, Calinski-Harabasz Index, and SVM classification on the test set (Rousseeuw, 1987; Caliński & Harabasz, 1974; Cortes & Vapnik, 1995; Zhan et al., 2020; Chhabra et al., 2021). Finally, to account for three main properties of disentangled representations, namely, modularity, compactness, and explicitness for the XYRCS and dSprites datasets, we have used eight supervised metrics: **z-diff**, **z-min** and **irs** from the intervention-based, **dci** and **sap** from the predictor-based, and **mig**, **jemmig** and **dcimig** from the information-based metric classes. The visual metrics can be used to demonstrate how the model aligns images and learns independent features. To avoid random seeds and training convergence affecting reliable comparison of these models as outlined in Locatello et al. (2019), we run all the models across all six data sets for ten different random seeds, reporting the best sought performance.

Furthermore, the proposed model offers remarkable training and inference performance by eliminating the need for a spatial decoder-based design. Table 6 in Appendix shows the training and inference performance of all the models across three different GPU architectures, namely, $V100$, $A100$ and $H100$. The results show that the training and inference performance of the proposed model are at least $10\times$, and $9\times$ compared to the average performance of other models, respectively.

Given the space constraints, we highlight the prominent results in the main part of the paper, while providing the remaining set of results and relevant material as part of the Appendix A.

### 4.1 NUMERICAL RESULTS

Here, we present the ability of the models to disentangle orientation information for the 5HDB dataset (Table 1), ability to learn translational and rotational invariant features for the MNISTU dataset (Table 2), and ability to disentangle features, including centroid and orientation, using the XYRCS and dSprites datasets (Tables 3, and 4). In the 5HDB dataset, we set a two-dimensional latent space in the DAE and $\beta$-VAE, and one-dimensional latent space in the other models because the other models explicitly learn $(x, y)$-positions, and rotation. We then use one feature, representing

Table 1: Comparison of models in their capability to learn orientations for the 5HDB test dataset, using the Pearson Correlation Coefficient (PCC, higher the better).

| Metrics | | Models | | | | | |
|---|---|---|---|---|---|---|---|
| | | CODAE | DAE | $\beta$-VAE | Spatial-VAE | Target-VAE | IRL-INR |
| PCC-Score | | **0.999** | 0.979 | 0.903 | 0.995 | 0.998 | **0.999** |

Table 2: Comparison of models on downstream task performance for the MNISTU dataset (Higher values better across all metrics).

| Metrics | | Models | | | | | |
|---|---|---|---|---|---|---|---|
| | | CODAE | DAE | $\beta$-VAE | Spatial-VAE | Target-VAE | IRL-INR |
| Silhouette Score | | **0.197** | $-0.020$ | $-0.034$ | 0.022 | 0.142 | 0.094 |
| Calinski-Harabasz Index | | **11186** | 3579 | 1511 | 1889 | 6077 | 9412 |
| SVM Classification | | 0.753 | 0.365 | 0.407 | 0.642 | **0.763** | 0.665 |

Table 3: Disentanglement scores for the XYRCS datasets. A higher value is preferred across all metric.

| Features | Models | Disentanglement scores | | | | | | | | |
|---|---|---|---|---|---|---|---|---|---|---|
| | | z-diff | z-var | irs | dci | sap | mig | jemmig | dcimig | avg |
| XYRCS | CODAE | **0.97** | 0.76 | 0.67 | **0.83** | 0.61 | **0.67** | **0.72** | **0.63** | **0.73** |
| | DAE | 0.89 | 0.80 | **0.87** | 0.66 | 0.63 | 0.48 | 0.64 | 0.53 | 0.68 |
| | $\beta$-VAE | 0.74 | 0.71 | 0.64 | 0.32 | 0.51 | 0.21 | 0.45 | 0.25 | 0.47 |
| | Spatial-VAE | 0.79 | 0.60 | 0.54 | 0.55 | 0.47 | 0.44 | 0.50 | 0.28 | 0.52 |
| | Target-VAE | 0.95 | **0.86** | 0.58 | 0.62 | **0.74** | 0.47 | 0.65 | 0.50 | 0.67 |
| | IRL-INR | 0.79 | 0.61 | 0.59 | 0.41 | 0.38 | 0.31 | 0.47 | 0.28 | 0.48 |
| CS | CODAE | **1.00** | **1.00** | **0.82** | **0.99** | **0.83** | **0.90** | 0.77 | **0.90** | **0.90** |
| | DAE | **1.00** | **1.00** | 0.68 | 0.77 | 0.48 | 0.64 | 0.64 | 0.72 | 0.74 |
| | $\beta$-VAE | 0.61 | **1.00** | 0.52 | 0.31 | 0.34 | 0.28 | 0.43 | 0.28 | 0.47 |
| | Spatial-VAE | **1.00** | **1.00** | 0.74 | 0.96 | 0.80 | 0.84 | **0.78** | 0.88 | 0.87 |
| | Target-VAE | **1.00** | **1.00** | 0.60 | 0.65 | 0.81 | 0.55 | 0.60 | 0.54 | 0.71 |
| | IRL-INR | 0.87 | 0.51 | 0.56 | 0.72 | 0.26 | 0.52 | 0.53 | 0.52 | 0.56 |

Table 4: Disentanglement scores for the dSprties datasets. A higher value is preferred across all metric.

| Features | Models | Disentanglement scores | | | | | | | | |
|---|---|---|---|---|---|---|---|---|---|---|
| | | z-diff | z-var | irs | dci | sap | mig | jemmig | dcimig | avg |
| XYRSS | CODAE | **0.97** | **0.85** | **0.82** | **0.56** | **0.59** | **0.48** | **0.64** | **0.45** | **0.67** |
| | DAE | 0.86 | 0.64 | 0.63 | 0.37 | 0.56 | 0.33 | 0.49 | 0.28 | 0.52 |
| | $\beta$-VAE | 0.80 | 0.69 | **0.82** | 0.50 | 0.57 | 0.36 | 0.57 | 0.36 | 0.58 |
| | Spatial-VAE | 0.67 | 0.48 | 0.57 | 0.32 | 0.31 | 0.20 | 0.37 | 0.15 | 0.38 |
| | Target-VAE | 0.88 | 0.78 | 0.54 | 0.36 | 0.54 | 0.37 | 0.51 | 0.32 | 0.53 |
| | IRL-INR | 0.89 | 0.72 | 0.69 | 0.45 | 0.45 | 0.28 | 0.51 | 0.25 | 0.53 |
| SS | CODAE | **1.00** | 0.99 | **0.74** | **0.89** | **0.51** | **0.58** | **0.71** | **0.64** | **0.75** |
| | DAE | 0.93 | **1.00** | 0.49 | 0.53 | 0.46 | 0.45 | 0.54 | 0.48 | 0.61 |
| | $\beta$-VAE | 0.94 | 0.46 | 0.47 | 0.51 | 0.49 | 0.53 | 0.57 | 0.57 | 0.56 |
| | Spatial-VAE | 0.92 | **1.00** | 0.56 | 0.53 | 0.50 | 0.42 | 0.55 | 0.45 | 0.61 |
| | Target-VAE | 0.90 | 0.82 | 0.54 | 0.46 | 0.41 | 0.48 | 0.55 | 0.42 | 0.57 |
| | IRL-INR | 0.99 | 0.99 | 0.65 | 0.45 | 0.44 | 0.35 | 0.59 | 0.40 | 0.60 |

the orientation to compute the PCC. The scores indicate that most models designed to learn orientation are capable of learning the rotations. However, the DAE and $\beta$-VAE, which are not designed to explicitly learn orientation, show sub-optimal performance compared to the other models. In the MNISTU, we use three metrics to assess the effectiveness of the models in downstream tasks using the learned latent space. Because the digit six and nine are symmetrical to each other, we combine them into one group. While the DAE and $\beta$-VAE have poor performance in the clustering and classifications, the CODAE, Target-VAE and IRL-INR efficiently learn features for the downstream task as shown in Table 2. Finally, we present the eight supervised disentanglement scores of the XYRCS and dSprites datasets in Table 3 and Table 4. In both datasets, we set a five-dimensional latent space in the DAE and $\beta$-VAE, and two-dimensional latent space in the other models for the same reason as in the 5HDB dataset. Firstly, all disentanglement scores are measured using all features. Then, invariant features are identified among five features, and the all scores are measured using their ground truth factors by excluding $(x, y)$ positions, and rotation factors. When considering the average of all metrics, the CODAE performs best for the XYRCS and dSprites datasets with and without $(x, y)$ positions, and rotation. For the XYRCS dataset, the Target-VAE performs as good as the CODAE

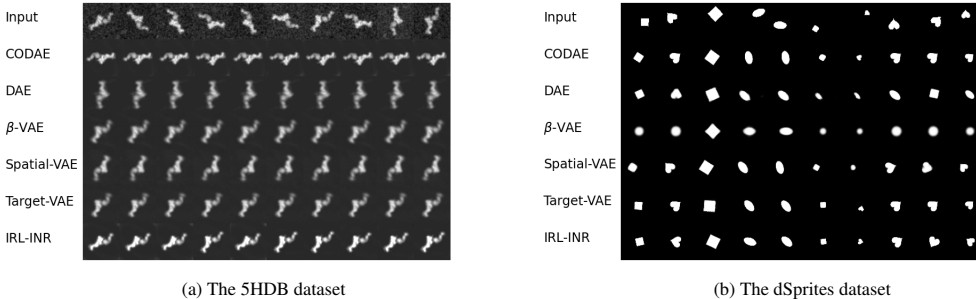

(a) The 5HDB dataset

(b) The dSprites dataset

Figure 2: Aligned reconstruction of the (a) 5HDB and (b) dSprites datasets. The first row represents the input images and the following rows represent the aligned reconstructed images by CODAE, DAE, $\beta$-VAE, Spatial-VAE, Target-VAE, and IRL-INR, respectively.

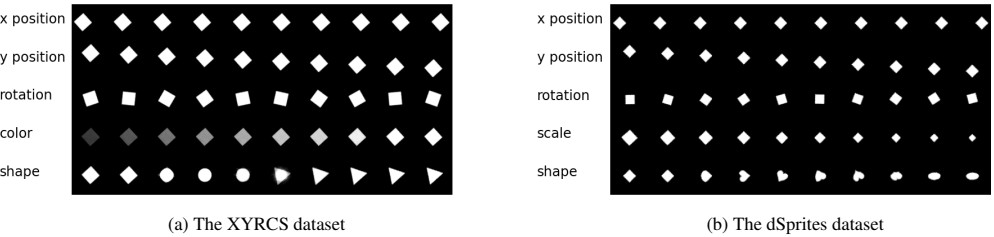

(a) The XYRCS dataset

(b) The dSprites dataset

Figure 3: Reconstructions of latent traversals across each latent dimension obtained by the CODAE for the (a) XYRCS and (b) dSprites datasets.

with all features while the Spatial-VAE exhibits a similar level performance to the CODAE in color and shape feature learning. For the dSprites dataset, the CODAE outperforms all models across all metrics when all features are considered. With only scale and shape features, it still outperforms the other models except **z-var** score. However, the difference from the best performance is marginal and negligible. Furthermore, when comparing the CODAE to those of the DAE, we can observe that disentanglement scores increase significantly with the translational and rotational equivariant layers.

## 4.2 Visual Results

We report the aligned reconstructions (for the 5HDB and dSprites datasets), and reconstructions of the latent traversals (for the XYRCS and dSprites datasets) in Figures 2 and 3, respectively. We repeat the same for the EMPIAR-10029 and Galaxy-zoo datasets in Figures 4 and Figures 5.

In the 5HDB dataset, the CODAE and IRL-INR are capable of reconstructing clearly visible and aligned images compared to the other models as shown in Figure 2. While the CODAE aligns images by preserving their scale and shape for the dSprites dataset, the other models fail to align heart images except the Target-VAE. Although the Target-VAE appears to align the inputs, the reconstruction of latent traversals in Figure 10 shows that it does not generate clear images. With EMPIAR-10029 and Galaxy-zoo datasets lacking ground truth factors, it is difficult to manually identify or verify the invariant features therein. For this reason, we present only the aligned reconstruction results for the CODAE, Spatial-VAE, Target-VAE, and IRL-INR models. It is clear that when the number of latent space dimensions is small, the IRL-INR model fails to reconstruct the inputs due to the reliance on the hypernetwork decoder, which demands large latent space dimension. For the Galaxy-zoo dataset, the CODAE is the only model that aligns input images with high quality, whereas the other models struggle to reconstruct when two galaxies are present in the input image as shown in Figure 4. Secondly, learned features by the CODAE for the XYRCS, dSprites, EMPIAR-10029 and Galaxy-zoo datasets are visualized in Figure 3 and Figure 5. Both figures show that the CODAE perfectly learn $(x, y)$ positions, and rotation. As a result, the CODAE disentangles color and shape in the XYRCS dataset, and scale and shape in the dSprites dataset. Furthermore, the CODAE enables discovering variations in lighting and shape in the EMPIAR-10029 dataset. For the Galaxy-zoo dataset, only the CODAE uncovers key features underpinning the dataset, such as size, color, shape, separation and background as shown in Figure 5 and Figrue 12. In addition to

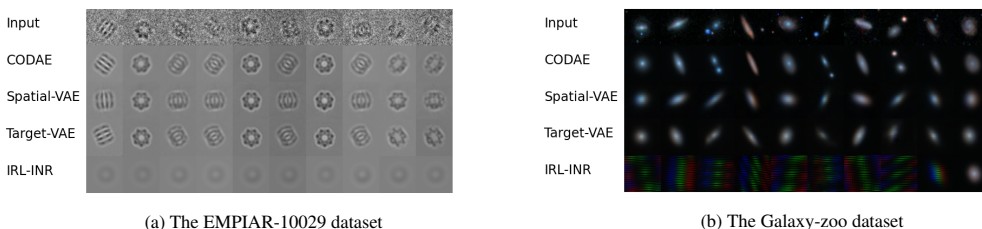

(a) The EMPIAR-10029 dataset

(b) The Galaxy-zoo dataset

Figure 4: Aligned reconstruction of the (a) EMPIAR-10029 and (b) Galaxy-zoo datasets. The first row represents the input images and the following rows represent the aligned reconstructed images by CODAE, Spatial-VAE, Target-VAE, and IRL-INR, respectively.

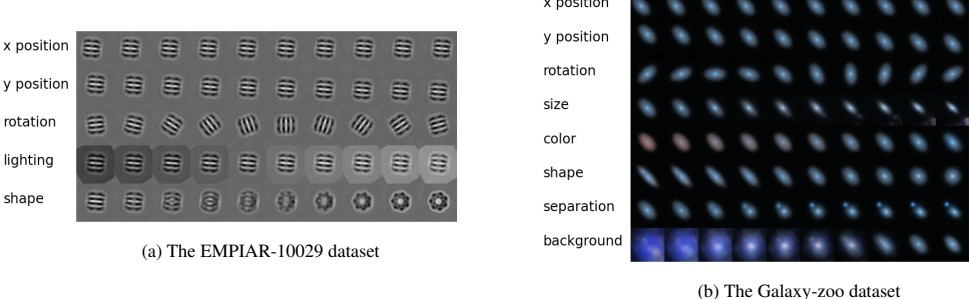

(a) The EMPIAR-10029 dataset

(b) The Galaxy-zoo dataset

Figure 5: Reconstructions of latent traversals across each latent dimension obtained by the CODAE for the (a) EMPIAR-10029 and (b) Galaxy-zoo datasets.

these, the latent spaces and their corresponding reconstructions of the MNISTU dataset, aligned reconstruction of the XYRCS dataset, and reconstructions of the latent traversals of all datasets across all models are shown in the Appendix A.

## 5    CONCLUSIONS

Learning distinct features, most importantly, centroids and orientations of shapes in two-dimensional images is crucially important for a number of image and video-processing applications, such as aligned image reconstructions, which has profound applications in various scientific domains. While the notion of translational and rotational equivariance and invariance in neural network-based models is very important, possessing these properties does not guarantee that such models learn centroids and orientations, especially along with other features.

In this paper, we proposed an approach that learns orthogonal features including centroids and orientations. By relying on translational and rotational equivariant layers and image moments, we presented a novel neural network architecture that can learn both equivariant and invariant features related to translation and rotation. Our evaluation, evaluated using six different datasets, including two realistic scientific datasets, against five different baseline models using both numerical and visual metrics, demonstrated that the proposed model can offer a superior performance for learning these two key features along with other features. More specifically, superior quality of the outputs for aligned reconstructions, and reconstructions of latent traversals across two scientific datasets showed that the full potential of the proposed approach. We are hopeful that the proposed model has real world utility in addressing a number of scientific problems.

Although the results are of superior quality, we believe there is room for further research. Specifically, we are interested in assessing the impact of image moments on noisy datasets, which leads to better understanding of the potential applications, albeit the fact that this is a separate study itself.

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

## A APPENDIX

### A.1 THE LATENT SPACES AND THEIR CORRESPONDING RECONSTRUCTIONS OF THE MNISTU DATASET

The latent spaces in the Figure 6 shows that DAE and $\beta$-VAE, which do not explicitly learn the rotation, are unable to group digits while the other models effectively group them. The Spatial-VAE is struggling to make one distinct group for the digit two, four, six, seven and nine. The IRL-INR also exhibits two clusters for the digit six and nine.

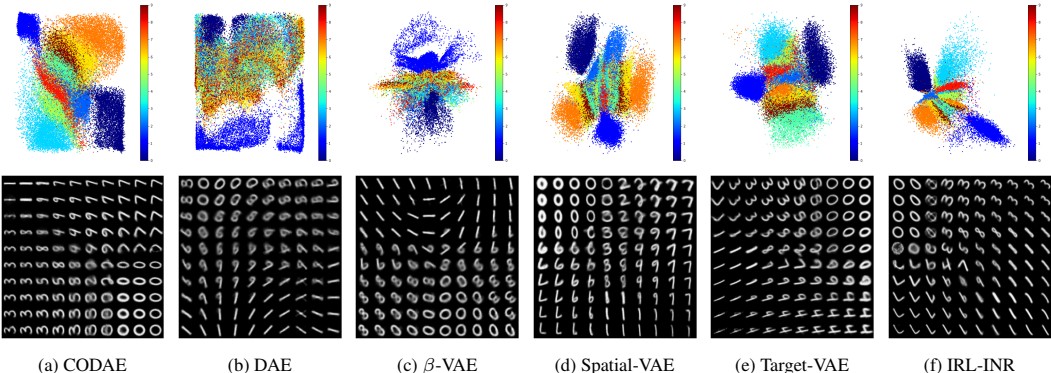

| (a) CODAE | (b) DAE | (c) $\beta$-VAE | (d) Spatial-VAE | (e) Target-VAE | (f) IRL-INR |

Figure 6: The latent spaces and their corresponding reconstruction of the MNISTU dataset when the dimension of the latent space is 2.

### A.2 THE ALIGNED RECONSTRUCTION OF THE XYRCS DATASET

We present the aligned reconstruction of the XYRCS dataset in the Appendix due to the page limit. The Figure 7 shows that DAE and $\beta$-VAE are struggling to align them while preserving the color and shape of the objects. Based on the disentanglement score in Table 3, the CODAE and Spatial-VAE perfectly align all objects while the Target-VAE is unable to align the squares and the IRL-INR faces difficulties to keep shapes when the object is either a circle or a rectangle.

### A.3 RECONSTRUCTIONS OF LATENT TRAVERSALS ACROSS EACH LATENT DIMENSION

The objects in the 5HDB dataset have two variations to their rigid body flexibility and e in-plane rotation. We use the e in-plane rotation feature to calculate the PC coefficient in Table 1. Therefore, we visualize only the rigid body flexibility in Figure 8. Once more, the DAE and $\beta$-VAE are struggling to reconstruct clear images. The CODAE exhibits the clear variation to rigid body flexibility. While the reconstructed images by both CODAE and IRL-INR are clear and sharp, there is a small variation to in-plane rotation in the outputs of IRL-INR.

The reconstructions of latent traversal across each latent dimension obtained by all models are shown in Figrue 9. While all models can disentangle $x$ position, $y$ position and color from the other features, DAE and $\beta$-VAE do not exhibit rotation using one feature. In this experiment, the Spatial-VAE disentangles all five factors while the Target-VAE fails to produce rectangles, making difficult to check

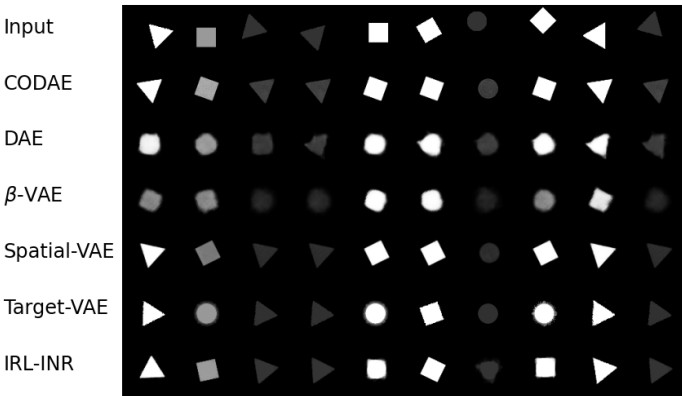

Figure 7: Aligned reconstruction of the XYRCS dataset. The first row represents the input images and the following rows represent the aligned reconstructed images by CODAE, DAE, $\beta$-VAE, Spatial-VAE, Target-VAE, and IRL-INR, respectively.

whether the model successfully learn all factors. The IRL-INR has another difficulty to separate the color and shape factor.

Similar to the XYRCS dataset, all models can disentangle $x$ position and $y$ position. However, CODAE and Spatial-VAE only seperate the scale and shape factors. Although the disentanglement scores of the $\beta$-VAE are relatively high in Table 4, its reconstruction performance is unsatisfactory as shown in Figure 2 and Figure 10, which is widely acknowledged by Burgess et al. (2018). Most models obtain lower disentanglement scores in the dSprties dataset than the XYRCS dataset, and the latent traversals also show the inabilities of each model to disentangle the dSprites dataset. The Spatial-VAE amd IRL-INR exhibit the rotation of the ellipse along with the shape variation while the Target-VAE is unable to reconstruct the clear images.

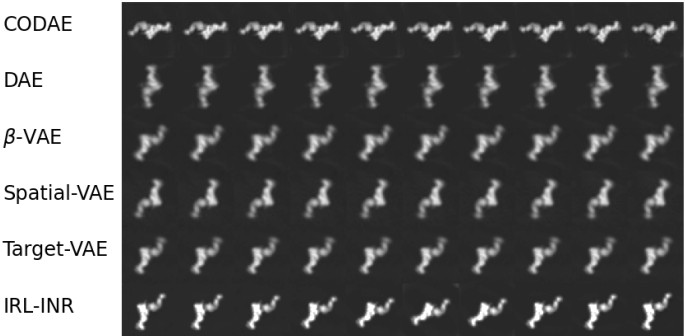

Figure 8: Reconstructions of latent traversal across the latent dimension obtained by CODAE, DAE, $\beta$-VAE, Spatial-VAE, Target-VAE, and IRL-INR for the 5HDB dataset.

### A.4 TRAINING DETAILS

To achieve the best numerical scores and reconstructions, all models are trained with four different learning rate: 0.001, 0.0005, 0.0001, and 0.00005 across all datasets. Each model is run for 100 epochs across all datasets except the Galaxy-zoo dataset. For the Galaxy-zoo dataset, the Spatial-VAE and Target-VAE are run for 200 epochs, and the IRL-INR is run for 300 epochs to guarantee their reconstruction losses are small enough for a comparison.

In all experiments, for $\beta$ in the loss function of the proposed model, we use four values: 1, 10, 50, and 100. The optimal beta is different from the datasets. Please see the Table 5 for the optimal beta

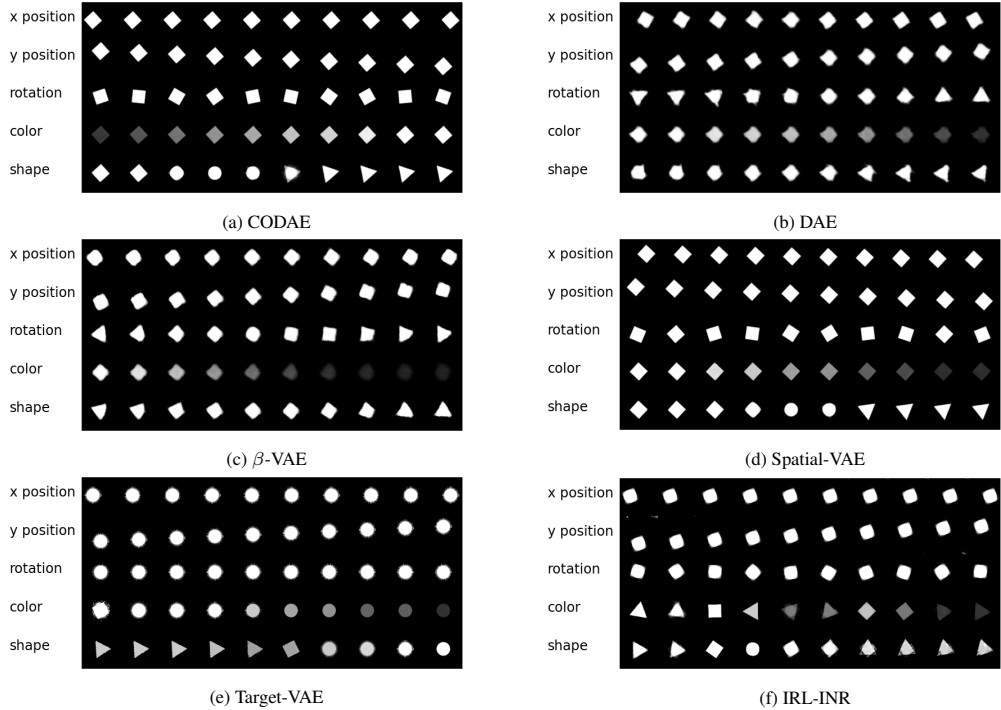

Figure 9: Reconstructions of latent traversal across each latent dimension obtained by CODAE, DAE, $\beta$-VAE, Spatial-VAE, Target-VAE, and IRL-INR for the XYRCS dataset.

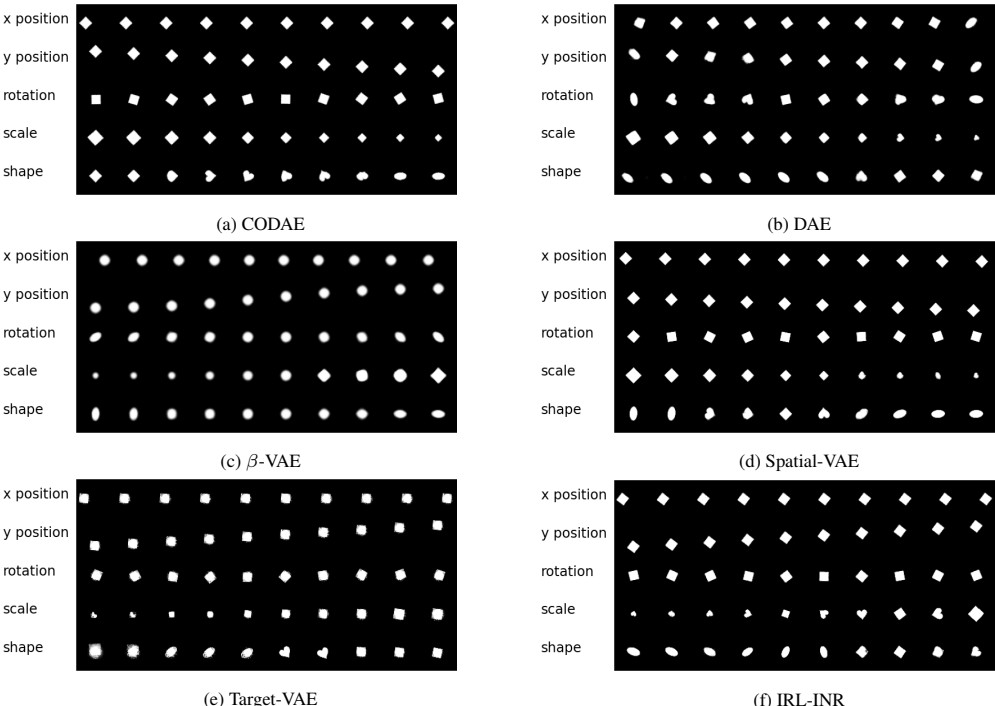

Figure 10: Reconstructions of latent traversal across each latent dimension obtained by CODAE, DAE, $\beta$-VAE, Spatial-VAE, Target-VAE, and IRL-INR for the dSprites dataset.

for each dataset. As previously discussed, $\beta$ is gradually decreased and reaches zero at a certain epoch. The final epochs for $\beta$ across the different datasets also can be found in the Table 5.

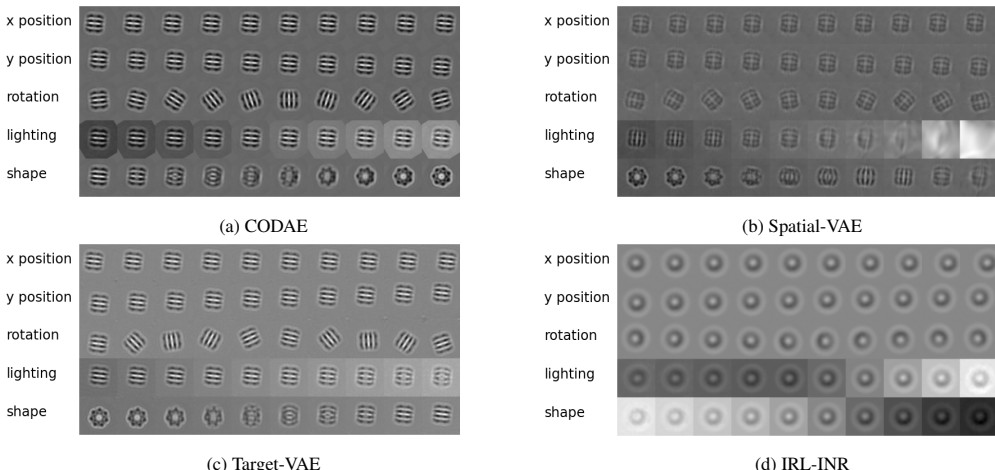

Figure 11: Reconstructions of latent traversal across each latent dimension obtained by CODAE, DAE, $\beta$-VAE, Spatial-VAE, Target-VAE, and IRL-INR for the EMPIAR-10029 dataset.

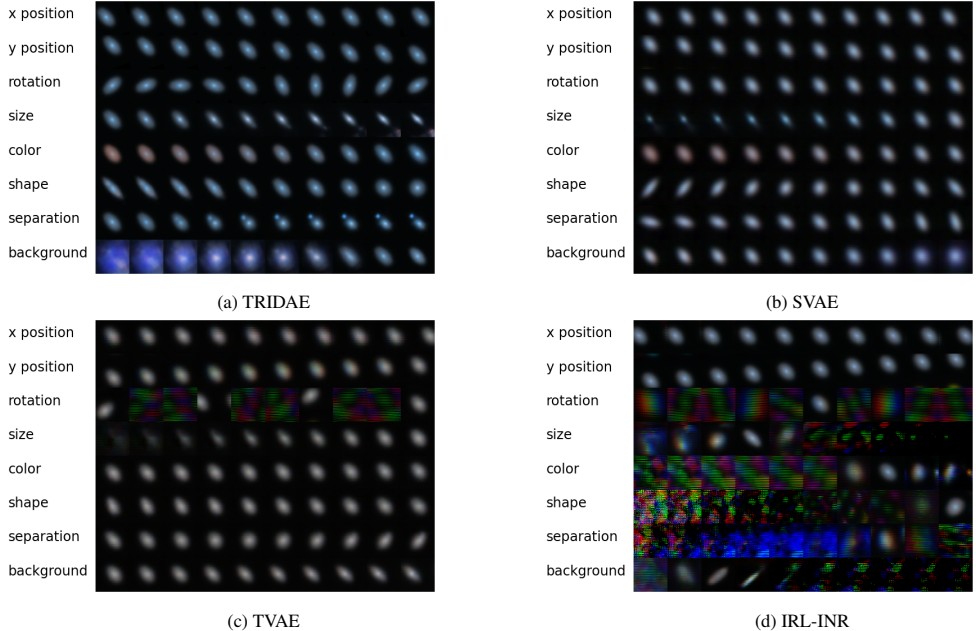

Figure 12: Reconstructions of latent traversal across each latent dimension obtained by CODAE, DAE, $\beta$-VAE, Spatial-VAE, Target-VAE, and IRL-INR for the Galaxy-zoo dataset.

## A.5 TRAINING AND INFERENCE SPEED ON DIFFERENT GPU MODELS

The proposed model uses the standard convolutional layers while the other centroid and orientation learning models use fully-connected layers or hypernetwork in their decoder parts. Therefore, the proposed model maintain the computational cost to the minimum. We verify this by measure the training and inference time on three different GPU architectures, namely, $V100$, $A100$ and $H100$. We provide the average of ten runs. The Table 6 shows that both training and inference speed of the other models is rapidly decreasing along with the dimension of images.

Table 5: The value $\beta$ used in CODAE for different datasets.

| Dataset | $\beta$ | Final epoch |
|---------|---------|-------------|
| 5HDB | 50 | 10 |
| MNISTU | 100 | 20 |
| XYRCS | 1 | 50 |
| dSprites | 1 | 10 |
| EMPIAR-10029 | 1 | 50 |
| Galaxy-zoo | 10 | 1 |

Table 6: Training and Inference speed per iteration on different GPU architectures. A higher value is preferred.

| Datasets | Models | V100 | | A100 | | H100 | |
|----------|--------|----------|-----------|----------|-----------|----------|-----------|
| | | training | inference | training | inference | training | inference |
| MNISTU | CODAE | 52.59 | 125.99 | 58.50 | 138.30 | 90.39 | 217.51 |
| ($28 \times 28 \times 1$) | Spatial-VAE | **81.04** | **151.40** | **140.29** | **256.73** | **189.50** | **450.85** |
| Batch size: 64 | Target-VAE | 20.14 | 43.15 | 37.16 | 58.72 | 48.75 | 95.88 |
| | IRL-INR | 2.83 | 11.52 | 4.05 | 13.70 | 5.77 | 23.51 |
| 5HDB | CODAE | **19.04** | 24.66 | **25.19** | **34.57** | **42.98** | 60.87 |
| ($40 \times 40 \times 1$) | Spatial-VAE | 2.77 | 6.56 | 5.60 | 14.03 | 5.37 | 13.27 |
| Batch size: 64 | Target-VAE | 13.09 | **28.13** | 24.76 | 30.64 | 30.23 | **67.09** |
| | IRL-INR | 2.37 | 10.55 | 3.63 | 12.62 | 5.16 | 23.24 |
| dSprites | CODAE | **37.26** | **86.75** | 41.76 | 99.27 | **63.67** | **135.35** |
| ($64 \times 64 \times 1$) | Spatial-VAE | 19.64 | 48.00 | **58.45** | **116.13** | 48.32 | 128.84 |
| Batch size: 64 | Target-VAE | 3.73 | 10.12 | 8.65 | 19.36 | 9.88 | 26.55 |
| | IRL-INR | 1.67 | 9.32 | 2.78 | 13.51 | 3.82 | 21.29 |
| XYRCS | CODAE | **43.24** | **98.55** | **45.67** | **99.61** | **66.53** | **134.01** |
| ($84 \times 84 \times 1$) | Spatial-VAE | 11.50 | 28.75 | 35.36 | 74.62 | 27.50 | 71.92 |
| Batch size: 64 | Target-VAE | 2.02 | 4.21 | 4.21 | 8.07 | 4.29 | 11.61 |
| | IRL-INR | 1.16 | 6.66 | 2.08 | 11.84 | 2.84 | 17.78 |
| Galaxy-zoo | CODAE | **22.01** | **55.42** | **29.56** | **61.68** | **48.35** | **99.62** |
| ($64 \times 64 \times 3$) | Spatial-VAE | 1.08 | 2.47 | 1.74 | 4.25 | 1.82 | 4.57 |
| Batch size: 64 | Target-VAE | 3.67 | 8.59 | 7.54 | 14.26 | 6.14 | 10.48 |
| | IRL-INR | 1.67 | 9.23 | 2.76 | 13.05 | 6.79 | 20.42 |
| EMPIAR-10029 | CODAE | **20.43** | **58.46** | **25.79** | **57.78** | **40.19** | **124.82** |
| ($128 \times 128 \times 1$) | Spatial-VAE | 1.76 | 1.25 | 1.06 | 2.21 | 1.05 | 2.27 |
| Batch size: 32 | Target-VAE | 1.21 | 2.27 | 3.02 | 4.08 | 3.00 | 7.56 |
| | IRL-INR | 1.68 | 7.55 | 3.23 | 14.29 | 4.24 | 20.12 |