# OpenReview forum: "Centroid- and Orientation-aware Feature Learning"
_ICLR.cc/2024/Conference — Submitted to ICLR 2024_

### Official Review · Reviewer_sBsU · 2023-10-29

**Soundness:** 2 fair
**Presentation:** 2 fair
**Contribution:** 2 fair
**Rating:** 3
**Confidence:** 4

**Summary:**

The paper proposes a method for learning representations where image moments (centroid and orientation) are explicitly disentangled from the rest of the representation. A loss term comparing the image moments is introduced and its contribution is gradually decreased during learning.

Experimental results on six datasets are provided to show that the proposed method compares favorably to six recently proposed methods that also seek to disentangle translation and rotation representations.

**Strengths:**

**S1.** The proposed method compares favorably to recently proposed methods. In particular, translation and rotation are effectively disentangled while the model is more computationally efficient than most other baselines.

**Weaknesses:**

**W1.** In general, the presentation could be significantly improved. Some examples:
- It seems to me that the key contribution of the method is the introduction of loss L_m but the discussion does not make this clear.
- Along the same lines, it is suggested (end of section 2.3) that the “primary focus” of the paper might be to achieve disentanglement (and indeed, some experiments also suggest that). However, it is not clear how this is pursued besides obtaining moments and orientation.
- The theorems in section 3.1 are barely referenced in subsequent sections.
- The experimental results need further details and discussion (more on this below).
- Some sentences are hard to understand, e.g.: in the abstract “training and inference performance” (what is the metric?), third-to-last sentence in paragraph preceding eq. (13) (on “subtle inaccuracy”).

**W2.** The motivation seems disconnected from the experimental validation. It is stated that learning of centroids and orientations “underpins” a number of downstream tasks. It is not clear what this means nor is it clear what the level of success of the proposed approach would be in this regard.

The downstream task experiments are perhaps most interesting but barely any details of the experimental setup are provided. A lot of space is taken by visual results but I would suggest the downstream task results are much more important.

**W3.** It is unclear what tables 3 and 4 convey when comparing different models as the optimal latent dimension is model dependent. For instance, for models TARGET-VAE and IRL-INR the original authors showed results for d >=32 (but it is suggested d=2 in the experiments in the present submission).

**Questions:**

**Q1.** Is the method pursuing disentanglement of all features or mainly/only obtaining moments and orientation? In case of the former, I would say this is not clear in the presentation, could you outline how this is pursued?

**Q2.** Why are baselines not compared with representation dimension as in the original papers?

---

> ### Author Response · Authors · 2023-11-16
> **Official Comment by Authors - Thank you for the review**
>
> We appreciate the time you've dedicated to reviewing our paper. Please find below our detailed response to your comments:
>
> 1. **Improvement of our paper (W1)**
>
> * We apologise for the omission of this proof since translational equivariance in CNN is a well-known concept in the community, and the proof is similar to the Theorem 3.2. We have now added the proof of theorem 3.1 in the Appendix.
>
> * Related to ‘training and inference performance’: We apologise for the confusion. It refers to the training and inference time. We have changed it in the paper.
>
> * Related to L_m loss: We highlight that
> > Although image moments capture the centroid and orientation features with a high degree of accuracy, they can still have a degree of subtle inaccuracy. Therefore, we utilize the image moments as guiding principles during the initial stages of learning.
>
>
>
> 2. **Problem statement and motivation of our paper (W2)**
>
> Thank you for the comments. We apologize if the motivation seems disconnected from the experimental validation. We have changed the first paragraph of the introduction to clarify our problem as follows:
>
> ‘The distinct 2D projections that arise from the rotation of a 3D object can be seen as unique categories of in-plane rotation. Each category corresponds to a specific orientation in the 3D space. This principle is widely applied in fields such as computer graphics, computer vision, and related areas where the understanding and manipulation of 3D objects in a 2D images are essential. In this context, robust learning of centroids and orientations of objects in images, along with other features, is important, especially when objects are captured with 3D shifts and rotations such as protein, and galaxy studies.’
>
> Therefore, we demonstrate that the capability of the model to learn key features, including centroids and orientations in the experimental validation. Table 3 and 4 show that DAE and beta-VAE have poor performance in factorizing two datasets, XYRCS and dSprites. On the other hand, the disentanglement scores of the proposed model on these datasets demonstrate that the model effectively learn all features including centroids and orientations.
>
>
> 3. **Optimal latent dimension for TARGET-VAE and IRL-INR (W3, Q2)**
>
> Target-VAE and IRL-INR conducted their experiments with large latent dimension, pursuing two objectives: 1) learning centroids and orientation, and 2) reconstructing the original data. Therefore, the two methods do not address the disentanglement scores in their paper, while they included the centroids and the orientation correlation. Therefore, the latent dimensions chosen in their respective papers are optimal for reconstructing the original data but may not effectively factorize all features. In contrast, our goal is not only to learn centroids and orientation but also to independently learn the other features while successfully reconstructing the original data. To show this, we chose a smaller latent dimension because modularity (or independence) and compactness are important in disentanglement properties. Figure 2, 4, 5, and 7 along with Table 3 and 4 demonstrate the capability of the model to factorize all the features while reconstructing the original data.
>
>
> 4. **Objective of our paper (Q1)**
>
> We pursue disentanglement of all features. DAE already shows an ability to disentangle features. However, it fails factorizing all features when the data has a rotational feature. Therefore, we introduce a new architecture outlined in the paper with $L_{moment}$ to explicitly learn centroids and rotations, leading to fully factorized features.
>
> 5. **Contributions of our paper (W1)**
>
> Our contribution is not limited on $L_{moment}$. We:
> * Introduce a translational and rotational equivariant encoder.
> * Propose an architecture that eliminates computationally expensive decoder, such as spatial or hypernetwork decoders.
> * Drastically reduce computational costs while achieving superior disentanglement scores.
>
>
>
> 6. **Further discussion**
>
> Despite detailed problem statement, clear motivation and challenges around robustly learning centroid and orientations, especially in real images, theoretical framework, and extensive experimental results (using appropriate datasets), we received a score of three. We would greatly appreciate more detailed feedback that led to this score on these specific areas. Understanding these details will help us improve and refine our paper. Your insights are invaluable, and we are eager to make the necessary updates based on your feedback.

---

> > ### Author Response · Authors · 2023-11-17
> > **Official Comment by Authors - Additional results**
> >
> > Please find below additional evaluation results.
> >
> > 1. **Ablation study**
> >
> > Following our earlier response, our contribution is not limited on $L_{moment}$. We have also Introduced a translational and rotational equivariant encoder. We have conducted the ablation study on the XYRCS and dSprites datasets without $L_{moment}$ or the translational and rotational equivariant encoder (TREE). The results show that both $L_{moment}$ and TREE are essential to achieve improved disentanglement scores. CODAE is a combination of DAE, TREE, and $L_{moment}$.
> >
> > * **XYRCS dataset**
> > | Features | Models | z-diff | z-var | irs | dci | sap | mig | jemmig | dcimig | avg |
> > |---|---|---|---|---|---|---|---|---|---|---|
> > | XYRCS | **CODAE** | **0.97** | **0.76** | 0.67 | **0.83** | 0.61 | **0.67** | **0.72** | **0.63** | **0.73** |
> > |  | DAE+TREE| 0.82 | 0.73 | 0.59 | 0.48 | 0.37 | 0.22 | 0.48 | 0.24 | 0.49 |
> > |  | DAE+$L_{moment}$ | 0.86 | 0.74 | **0.87** | 0.57 | **0.63** | 0.44 | 0.61 | 0.51 | 0.65 |
> > | CS | **CODAE** | **1.00** | **1.00** | **0.82** | **0.99** | **0.83** | **0.90** | **0.77** | **0.90** | **0.90** |
> > |  | DAE+TREE| 0.92 | 0.81 | 0.38 | 0.35 | 0.13 | 0.16 | 0.47 | 0.16 | 0.42 |
> > |  | DAE+$L_{moment}$ | 0.57 | 0.78 | 0.45 | 0.20 | 0.00 | 0.01 | 0.29 | 0.01 | 0.28 |
> >
> >
> >
> >
> > * **dSprites dataset**
> > | Features | Models | z-diff | z-var | irs | dci | sap | mig | jemmig | dcimig | avg |
> > |---|---|---|---|---|---|---|---|---|---|---|
> > | XYRCS | **CODAE** | **0.97** | **0.85** | **0.82** | **0.56** | **0.59** | **0.48** | **0.64** | **0.45** | **0.67** |
> > |  | DAE+TREE| 0.92 | 0.78 | 0.65 | 0.36 | 0.34 | 0.30 | 0.45 | 0.21 | 0.50 |
> > |  | DAE+$L_{moment}$ | 0.71 | 0.56 | 0.42 | 0.24 | 0.33 | 0.19 | 0.37 | 0.14 | 0.37 |
> > | SS | **CODAE** | **1.00** | **0.99** | **0.74** | **0.89** | **0.51** | **0.58** | **0.71** | **0.64** | **0.75** |
> > |  | DAE+TREE| 0.95 | 0.83 | 0.29 | 0.61 | 0.38 | 0.55 | 0.61 | 0.53 | 0.59 |
> > |  | DAE+$L_{moment}$ | 0.84 | 0.53 | 0.34 | 0.33 | 0.44 | 0.40 | 0.46 | 0.38 | 0.46 |
> >
> >
> > 2. **GF-Score**
> >
> > We have posted the GF-Score of two real datasets, namely EMPIAR-10029 (proteins) and Galaxy Zoo, that do not have labels. As specified in [1], a lower value is preferred. Additionally, it's important to note that the EMPIAR-10029 dataset comprises  distinct 2D projections resulting from the rotation of a 3D object.
> >
> > | GF-Score |  |  |
> > |---|---|---|
> > | Models/Datasets | EMPIAR-10029 | Galaxy-zoo |
> > | CODAE | **0.000039** | **0.000088** |
> > | Spatial-VAE | 0.000820 | 0.004332 |
> > | Target-VAE | 0.000766 | 0.002456 |
> > | IRL-INR | 0.000995 | 0.001182 |
> >
> >
> >
> > [1] Orthogonality-Enforced Latent Space in Autoencoders: An Approach to Learning Disentangled Representations, ICML2023

---

> > > ### Comment · Reviewer_sBsU · 2023-11-21
> > > **Response to authors**
> > >
> > > Thank you for the rebuttal and additional results.
> > >
> > > The rebuttal adds some clarity. I have spent an additional few hours trying to understand the contribution of the submission and still have several questions.
> > >
> > > Further, all reviewers seem aligned that the submission is difficult to understand. Yet, the rebuttal insists "detailed problem statement, clear motivation" were provided. This is dismissive of the reviewers' comments. I would discourage such interaction. In fact, even the new language provided in the rebuttal lacks clarity, e.g., "Each category corresponds to in the 3D space." -- this is not a proper sentence, nor is it clear.
> > >
> > > The rebuttal claims there are contributions besides L_m:
> > >
> > > 1. "Introduce a translational and rotational equivariant encoder."
> > >
> > > Unclear what is novel. The architecture is based on DAE. Interestingly the new results in the rebuttal show that the alleged contributions hurt DAE performance (e.g., compare the performance of DAE in the submission to the performances in the rebuttal).
> > >
> > > 2. "Propose an architecture that eliminates computationally expensive decoder, such as spatial or hypernetwork decoders."
> > >
> > > The decoder contribution would need to be presented in detail and experimental results included to make it clear what the achieved improvements are. I would also note that prior work, e.g., [Chat and Thiyagalingam ICML 2023] use the same decoder across baselines in the interest of a fair evaluation. The present submission would do well to incorporate evaluations with a constant decoder.
> > >
> > > 3. "Drastically reduce computational costs while achieving superior disentanglement scores."
> > >
> > > Unclear where the reduced computational costs would come from (besides the change in decoder).
> > >
> > > Given consideration to all of the above, I chose to keep my current score for the time being.

---

> ### Author Response · Authors · 2023-11-21
> **Response to the reviewer**
>
> Thank you for your response, and we appreciate that your views, and glad we added some clarity. Please find below our detailed response to your additional comments:
>
> 1. **Comments on ‘Each category corresponds to in the 3D space." -- this is not a proper sentence, nor is it clear’:**
>
> Thank you for your feedback, and we apologize for the missing word. The original sentence was:
> ‘The distinct 2D projections that arise from the rotation of a 3D object can be seen as unique categories of in-plane rotation. Each category corresponds to *a specific orientation* in the 3D space.’
>
> We have revised all the response. We truly appreciate your feedback.
>
> 2. **Introduce a translational and rotational equivariant encoder. (Q1)**
>
> We would like to emphasize that the primary objective of DAE is to disentangle all features. However, as illustrated in Table 3 and 4, and Figure 9 and 10, DAE has poor performance in factorizing all features when the data has a rotational feature. Additionally, partial modifications to DAE may negatively impact its overall performance. This forms the motivation for the proposed work, CODAE, where we introduce two aspects, which we believe underpins the novelty: (i) a translational and rotational equivariant encoder, and (ii) a loss function based on image moments.
> We are in agreement that neither the translational and rotational equivariant encoder nor image moments alone can improve the disentanglement scores, for the reasons of:
>
> * The translational and rotational equivariant encoder, designed following the Theorem 3.2, has the limitation in practice due to the impossibility of integration on S1. The computations are performed on a subset of S1, limiting rotational equivariance.
> * While image moments capture the centroid and orientation features with a high degree of accuracy, they can still have a degree of subtle inaccuracy.
>
> Therefore, we use the image moments as guiding principles during the initial stages of learning. Our evaluation results show that the combination of a translational and rotational equivariant encoder with image moments significantly improves disentanglement scores, particularly when the data involves rotational features. Figure 5 illustrates that the capability of the proposed architecture to learn semantic representations of objects when the objects of interest are positioned and oriented arbitrarily within the image.
>
> We will add these as part of the modified version of the manuscript to improve the clarity.
>
> 3. **Propose an architecture that eliminates computationally expensive decoder, such as spatial or hypernetwork decoders. (Q2)**
>
> While we understand the rationale for the suggestion, we politely would like to disagree with the reviewer’s comments mainly because different architectures rely on its own decoder design. More specifically:
>
> * The contribution of Spatial-VAE lies in its spatial decoder.
> * The contribution of Target-VAE is its specialized encoder, built on top of the spatial decoder from Spatial-VAE.
> * The contribution of IRL-INR is its hypernetwork decoder.
>
> As such, each model's distinctive feature is embedded in its decoder design. As such, it is almost impossible to rely on the same decoder unless we extensively modify their respective designs, distorting their purposes, which would be very unfair comparison. Given this, our method of comparison is fair retaining their design in the referenced papers [1, 2,3]. However, we used the same decoder for CODAE, DAE and $\beta$-VAE, which is valid and permissible.
>
>
> 4. **Drastically reduce computational costs while achieving superior disentanglement scores. (Q3)**
>
> Continuing from the previous response, each model has its own encoder or decoder model (as intended than modified or distorted) to learn semantic representations of objects when the objects of interest are positioned and oriented arbitrarily within the image. Our proposal in the paper involves designing a translational and rotational equivariant encoder while avoiding spatial or hypernetwork decoders. We believe this approach provides a clear rationale for reducing computational costs.
>
>
>
> [1] Explicitly disentangling image content from translation and rotation with spatial-vae, Neurips 2019
>
> [2] Unsupervised object representation learning using translation and rotation group equivariant vae, Neurips 2022
>
> [3] Rotation and translation invariant representation learning with implicit neural representations, ICML 2023

---

### Official Review · Reviewer_8vxB · 2023-11-01

**Soundness:** 1 poor
**Presentation:** 2 fair
**Contribution:** 1 poor
**Rating:** 3
**Confidence:** 3

**Summary:**

The paper proposes to disentangle the data into invariant and equivariant components. It builds up on DAE and introduces image moment based losses. The results are promising and the evaluation is extensive.

**Strengths:**

- The method is simple to understand and well-written.
- Evaluations are extensive for the disentanglement property.

**Weaknesses:**

- Due to the deterministic and simple nature of the moment computation, it could be easy for the neural network to learn z_eq. Therefore, a result on ablating L_{moment} could be interesting to see the emergent properties just based on the reconstruction loss, and also could be a baseline, since moment loss is the only new component here. As a corollary, the moment loss could also be applied over other baselines to evaluate how much does it contribute in improving their performance.
- Novelty of the moment loss is very limited, as it is a widely known concept in the community.
- Results on 3D datasets, such as 3D airplanes, 3D teapots, 3D face could test the method more robustly as the shape also changes.
- GF-Score could be reported as proposed in the DAE paper.

**Questions:**

- How is the moment computed for other factors such as shape and color? Is anything more than centroid and orientation that is part of z_{eq} on these datasets?

---

> ### Author Response · Authors · 2023-11-16
> **Official Comment by Authors - Thank you for the review**
>
> We appreciate the time you've dedicated to reviewing our paper. Please find below our detailed response to your comments:
>
> 1. **Ablation study (W)**
>
> We conduct the ablation study, and the results indicate that the disentanglement scores increase with the inclusion of the $L_{moment}$. We will include the ablation study table in the paper and post it separately in the response soon.
>
>
> 2. **Contributions of our paper (W)**
>
> While the concept of moment loss is widely recognized in the community, this paper introduces the first method to incorporate image moments into the learning mechanism -  which is not common. In addition, our contribution is not limited on $L_{moment}$. We:
> * Introduce a translational and rotational equivariant encoder.
> * Propose an architecture that eliminates computationally expensive decoder, such as spatial or hypernetwork decoders.
> * Drastically reduce computational costs while achieving superior disentanglement scores.
>
>
> 3. **Additional experiments (W)**
>
> Thank you for the comments. We have conducted additional experiments and new results will be posted separately soon.
>
>
> 4. **GF-Score (W)**
>
> Thank you for the comments. We have added the scores in the paper and will post it separately in the response soon.
>
>
> 5. **Image moments for other factors such as shape and color**
>
> We do not compute moments for other factors such as shape and color. Based on disentanglement literature (beta-VAE and DAE), these features can be learned. However, the table 3 and 4 show that these models struggle to learn features like shape and color when objects are captured with orientations. Therefore, the primary object of this paper is to guide the network to learn centroids and orientations using image moments, leading to the proposed model to effectively learn other features as part of $z_{inv}$.
>
>
> 6. **Further discussion**
>
>  Despite detailed problem statement, clear motivation and challenges around robustly learning centroid and orientations, especially in real images, theoretical framework, and extensive experimental results (using appropriate datasets), we received a score of three. We would greatly appreciate more detailed feedback that led to this score on these specific areas. Understanding these details will help us improve and refine our paper. Your insights are invaluable, and we are eager to make the necessary updates based on your feedback.

---

> > ### Author Response · Authors · 2023-11-17
> > **Official Comment by Authors - Additional results**
> >
> > Please find below additional evaluation results.
> >
> > 1. **Ablation study**
> >
> > Following our earlier response, our contribution is not limited on $L_{moment}$. We have also Introduced a translational and rotational equivariant encoder. We have conducted the ablation study on the XYRCS and dSprites datasets without $L_{moment}$ or the translational and rotational equivariant encoder (TREE). The results show that both $L_{moment}$ and TREE are essential to achieve improved disentanglement scores. CODAE is a combination of DAE, TREE, and $L_{moment}$.
> >
> > * **XYRCS dataset**
> > | Features | Models | z-diff | z-var | irs | dci | sap | mig | jemmig | dcimig | avg |
> > |---|---|---|---|---|---|---|---|---|---|---|
> > | XYRCS | **CODAE** | **0.97** | **0.76** | 0.67 | **0.83** | 0.61 | **0.67** | **0.72** | **0.63** | **0.73** |
> > |  | DAE+TREE| 0.82 | 0.73 | 0.59 | 0.48 | 0.37 | 0.22 | 0.48 | 0.24 | 0.49 |
> > |  | DAE+$L_{moment}$ | 0.86 | 0.74 | **0.87** | 0.57 | **0.63** | 0.44 | 0.61 | 0.51 | 0.65 |
> > | CS | **CODAE** | **1.00** | **1.00** | **0.82** | **0.99** | **0.83** | **0.90** | **0.77** | **0.90** | **0.90** |
> > |  | DAE+TREE| 0.92 | 0.81 | 0.38 | 0.35 | 0.13 | 0.16 | 0.47 | 0.16 | 0.42 |
> > |  | DAE+$L_{moment}$ | 0.57 | 0.78 | 0.45 | 0.20 | 0.00 | 0.01 | 0.29 | 0.01 | 0.28 |
> >
> >
> >
> >
> > * **dSprites dataset**
> > | Features | Models | z-diff | z-var | irs | dci | sap | mig | jemmig | dcimig | avg |
> > |---|---|---|---|---|---|---|---|---|---|---|
> > | XYRCS | **CODAE** | **0.97** | **0.85** | **0.82** | **0.56** | **0.59** | **0.48** | **0.64** | **0.45** | **0.67** |
> > |  | DAE+TREE| 0.92 | 0.78 | 0.65 | 0.36 | 0.34 | 0.30 | 0.45 | 0.21 | 0.50 |
> > |  | DAE+$L_{moment}$ | 0.71 | 0.56 | 0.42 | 0.24 | 0.33 | 0.19 | 0.37 | 0.14 | 0.37 |
> > | SS | **CODAE** | **1.00** | **0.99** | **0.74** | **0.89** | **0.51** | **0.58** | **0.71** | **0.64** | **0.75** |
> > |  | DAE+TREE| 0.95 | 0.83 | 0.29 | 0.61 | 0.38 | 0.55 | 0.61 | 0.53 | 0.59 |
> > |  | DAE+$L_{moment}$ | 0.84 | 0.53 | 0.34 | 0.33 | 0.44 | 0.40 | 0.46 | 0.38 | 0.46 |
> >
> >
> > 2. **GF-Score**
> >
> > We have posted the GF-Score of two real datasets, namely EMPIAR-10029 (proteins) and Galaxy Zoo, that do not have labels. As specified in [1], a lower value is preferred. Additionally, it's important to note that the EMPIAR-10029 dataset comprises  distinct 2D projections resulting from the rotation of a 3D object.
> >
> > | GF-Score |  |  |
> > |---|---|---|
> > | Models/Datasets | EMPIAR-10029 | Galaxy-zoo |
> > | CODAE | **0.000039** | **0.000088** |
> > | Spatial-VAE | 0.000820 | 0.004332 |
> > | Target-VAE | 0.000766 | 0.002456 |
> > | IRL-INR | 0.000995 | 0.001182 |
> >
> >
> >
> > [1] Orthogonality-Enforced Latent Space in Autoencoders: An Approach to Learning Disentangled Representations, ICML2023

---

### Official Review · Reviewer_ULgT · 2023-11-03

**Soundness:** 2 fair
**Presentation:** 3 good
**Contribution:** 2 fair
**Rating:** 3
**Confidence:** 4

**Summary:**

This paper proposes to learn image representations with translational and rotational invariance and equivariance properties, under the guidance of the centroid and orientation information of images.  The model is trained with simple image reconstruction loss in the space of pixel intensity and image moments.  Experiments on several datasets (such as 5HD and MINIST) demonstrate that the proposed method outperforms existing methods.

**Strengths:**

1. It is technically reasonable to guide the learning of equivariant features with some spatial image statistics (such as image centroid).
2. The paper is well-organized and easy to follow.
3. The performance across multiple benchmarks consistently shows the improvement of the proposed method over existing works.

**Weaknesses:**

The main technical contribution of this work is to guide the learning of equivariant features with some spatial image statistics (such as image centroid). However, all the experiments are conducted on toy datasets such as MNIST digits which contain very simple 2D objects and almost uniform background region. This is also manifested in the evaluation scores. For example, in Table 1, all methods achieve over 97% accuracy. This leaves a question mark on how useful the proposed method is in practice where natural images are way more complicated and whether the simple spatial statistics are still sufficient.

**Questions:**

How is the performance of the proposed method on natural image datasets, such as CIFAR or ImageNet where objects and background are more complicated and rotations are almost 3D (instead of just 2D in-plane rotation)?

---

> ### Author Response · Authors · 2023-11-16
> **Official Comment by Authors - Thank you for the review**
>
> Thank you for your comments and questions.
>
> 1. **All the experiments are conducted on toy datasets such as MNIST digits which contain very simple 2D objects and almost uniform background region (W).**
>
> First of all, we would like to politely highlight that the statement “all the experiments are conducted on toy datasets such as MNIST digits which contain very simple 2D objects and almost uniform background region.” Undermined all the efforts and work presented in this paper. In contrary to the statement, the evaluation includes two real world datasets, such as EMPIAR-10029 (protein) and Galaxy-Zoo datasets, where objects and background are complicated, and hence do not have a uniform background. Figures 4 and 5 clearly show that our proposed model learns key features of complicated datasets including centroids and orientations. Secondly, aside from Table 1, the disentanglement scores in Table 3 and 4 are far better than those presented in the literature and those achieved by the baselines presented in the paper. As such, we find the dismissal not only disheartening but also expects an unrealistic accuracy on these tasks. Please see our detailed responses below:
>
>
> * However, all the experiments are conducted on toy datasets such as MNIST digits which contain very simple 2D objects and almost uniform background region.
>
> > Please see the opening response above. Evaluations include rather complex EMPIAR and GalaxyZoo datasets. In particular the EMPIAR (10029) dataset includes 3D rotation.
>
>
> * How is the performance of the proposed method on natural image datasets, such as CIFAR or ImageNet…..
>
> > The primary objective of the proposed work is to design a method that learns key features including centroid and orientation. Our selection of datasets is based on the literature and on existing work on disentanglement methods, but now includes additional 3D datasets as requested by the Reviewer 8vxB (3D datasets, such as 3D airplanes, 3D teapots, 3D face). As such, including CIFAR or ImageNet is beyond the scope of disentanglement studies.
>
>
> 2. **Problem statement and motivation of our paper**
>
> We agree that we could have phrased the problem statement much more clearly, which, based on the comments, rephrased as follows:
> ‘The distinct 2D projections that arise from the rotation of a 3D object can be seen as unique categories of in-plane rotation. Each category corresponds to a specific orientation in the 3D space. This principle is widely applied in fields such as computer graphics, computer vision, and related areas where the understanding and manipulation of 3D objects in 2D images are essential. In this context, robust learning of centroids and orientations of objects in images, along with other features, is important, especially when objects are captured with 3D shifts and rotations such as protein, and galaxy studies.’
>
>
> 3. **Further discussion**
>
>  We feel that the score of three (3) is purely based on misunderstanding of the difficulty of the challenges around robust learning of the centroids and orientations (please see [1, 2, 3]), our evaluation process (such as missing the fact that we included real-world and complex datasets) or overlooking the comparison in Table 3 and 4. As such, we feel that the lower scores are not justified.  We believe additional datasets, and more extensive evaluation, and our responses above clear these issues. We will be happy to receive more detailed feedback that led to this score on these specific areas.
>
> [1] Explicitly disentangling image content from translation and rotation with spatial-vae, Neurips 2019
>
> [2] Unsupervised object representation learning using translation and rotation group equivariant vae, Neurips 2022
>
> [3] Rotation and translation invariant representation learning with implicit neural representations, ICML 2023

---

> > ### Author Response · Authors · 2023-11-17
> > **Official Comment by Authors - Additional results**
> >
> > Please find below additional evaluation results.
> >
> > 1. **Ablation study**
> >
> > Following our earlier response, our contribution is not limited on $L_{moment}$. We have also Introduced a translational and rotational equivariant encoder. We have conducted the ablation study on the XYRCS and dSprites datasets without $L_{moment}$ or the translational and rotational equivariant encoder (TREE). The results show that both $L_{moment}$ and TREE are essential to achieve improved disentanglement scores. CODAE is a combination of DAE, TREE, and $L_{moment}$.
> >
> > * **XYRCS dataset**
> > | Features | Models | z-diff | z-var | irs | dci | sap | mig | jemmig | dcimig | avg |
> > |---|---|---|---|---|---|---|---|---|---|---|
> > | XYRCS | **CODAE** | **0.97** | **0.76** | 0.67 | **0.83** | 0.61 | **0.67** | **0.72** | **0.63** | **0.73** |
> > |  | DAE+TREE| 0.82 | 0.73 | 0.59 | 0.48 | 0.37 | 0.22 | 0.48 | 0.24 | 0.49 |
> > |  | DAE+$L_{moment}$ | 0.86 | 0.74 | **0.87** | 0.57 | **0.63** | 0.44 | 0.61 | 0.51 | 0.65 |
> > | CS | **CODAE** | **1.00** | **1.00** | **0.82** | **0.99** | **0.83** | **0.90** | **0.77** | **0.90** | **0.90** |
> > |  | DAE+TREE| 0.92 | 0.81 | 0.38 | 0.35 | 0.13 | 0.16 | 0.47 | 0.16 | 0.42 |
> > |  | DAE+$L_{moment}$ | 0.57 | 0.78 | 0.45 | 0.20 | 0.00 | 0.01 | 0.29 | 0.01 | 0.28 |
> >
> >
> >
> >
> > * **dSprites dataset**
> > | Features | Models | z-diff | z-var | irs | dci | sap | mig | jemmig | dcimig | avg |
> > |---|---|---|---|---|---|---|---|---|---|---|
> > | XYRCS | **CODAE** | **0.97** | **0.85** | **0.82** | **0.56** | **0.59** | **0.48** | **0.64** | **0.45** | **0.67** |
> > |  | DAE+TREE| 0.92 | 0.78 | 0.65 | 0.36 | 0.34 | 0.30 | 0.45 | 0.21 | 0.50 |
> > |  | DAE+$L_{moment}$ | 0.71 | 0.56 | 0.42 | 0.24 | 0.33 | 0.19 | 0.37 | 0.14 | 0.37 |
> > | SS | **CODAE** | **1.00** | **0.99** | **0.74** | **0.89** | **0.51** | **0.58** | **0.71** | **0.64** | **0.75** |
> > |  | DAE+TREE| 0.95 | 0.83 | 0.29 | 0.61 | 0.38 | 0.55 | 0.61 | 0.53 | 0.59 |
> > |  | DAE+$L_{moment}$ | 0.84 | 0.53 | 0.34 | 0.33 | 0.44 | 0.40 | 0.46 | 0.38 | 0.46 |
> >
> >
> > 2. **GF-Score**
> >
> > We have posted the GF-Score of two real datasets, namely EMPIAR-10029 (proteins) and Galaxy Zoo, that do not have labels. As specified in [1], a lower value is preferred. Additionally, it's important to note that the EMPIAR-10029 dataset comprises  distinct 2D projections resulting from the rotation of a 3D object.
> >
> > | GF-Score |  |  |
> > |---|---|---|
> > | Models/Datasets | EMPIAR-10029 | Galaxy-zoo |
> > | CODAE | **0.000039** | **0.000088** |
> > | Spatial-VAE | 0.000820 | 0.004332 |
> > | Target-VAE | 0.000766 | 0.002456 |
> > | IRL-INR | 0.000995 | 0.001182 |
> >
> >
> >
> > [1] Orthogonality-Enforced Latent Space in Autoencoders: An Approach to Learning Disentangled Representations, ICML2023

---

### Meta-Review · Area_Chair_kkog · 2023-12-09

**Metareview:**

An extension of the disentangling autoencoder is proposed through unsupervised learning of centroid and orientation information. The paper received three reviews, all of which recommend to reject. The AC carefully went through the paper, reviews and author feedback. A central issue for Reviewer ULgT is the choice of evaluation datasets, for which the rebuttal points out that choices made are similar to prior works on disentanglement. However, the central question is whether the family of methods including both the proposed method and prior ones like DAE have applicability to natural image datasets, which the author feedback does not address. An experiment could be conducted, or an argument made that disentanglement methods are not able to handle such data as yet. The experimental requirements raised by Reviewer 8vxB are sufficiently addressed by the author feedback. The confusion around the nature of the contribution being limited to L_moment is initially shared with Reviewer sBsU, but the author feedback establishes TREE as a contribution too. The necessary ablation of each component is provided in the rebuttal.

But limitations of each of the above two aspects, their necessity to work together and how these design choices influence training are only discussed briefly in response to a second round of questions. The AC agrees with the opinion of Reviewer sBsU that the positioning of these choices is not clear in the presentation of the paper and the paper will greatly benefit from a revision that discusses these in detail, augmented with all the additional information discussed in the author feedback. Overall, the paper presents potentially interesting new ideas, which can be understood better with presentation and experimentation that fully incorporate the review feedbacks for submission at a future venue, but may not be accepted at ICLR in current form.

**Justification For Why Not Higher Score:**

Potentially interesting new ideas are presented, albeit not fully analyzed in discussion and experimentation.

**Justification For Why Not Lower Score:**

Not applicable.

---

### Decision · Program_Chairs · 2024-01-16

Reject